# CamCo: Camera-Controllable 3D-Consistent Image-to-Video Generation

## Abstract

Recently video diffusion models have emerged as expressive generative tools for high-quality video content creation readily available to general users. However, these models often do not offer precise control over camera poses for video generation, limiting the expression of cinematic language and user control. To address this issue, we introduce **CamCo**, which allows fine-grained **Cam**era pose **Co**ntrol for image-to-video generation. We equip a pre-trained image-to-video generator with accurately parameterized camera pose input using Plücker coordinates. To enhance 3D consistency in the videos produced, we integrate an epipolar attention module in each attention block that enforces epipolar constraints to the feature maps. Additionally, we fine-tune CamCo on real-world videos with camera poses estimated through structure-from-motion algorithms to better synthesize object motion. Our experiments show that CamCo significantly improves 3D consistency and camera control capabilities compared to previous models while effectively generating plausible object motion. Project page: https://camco2024.github.io/.

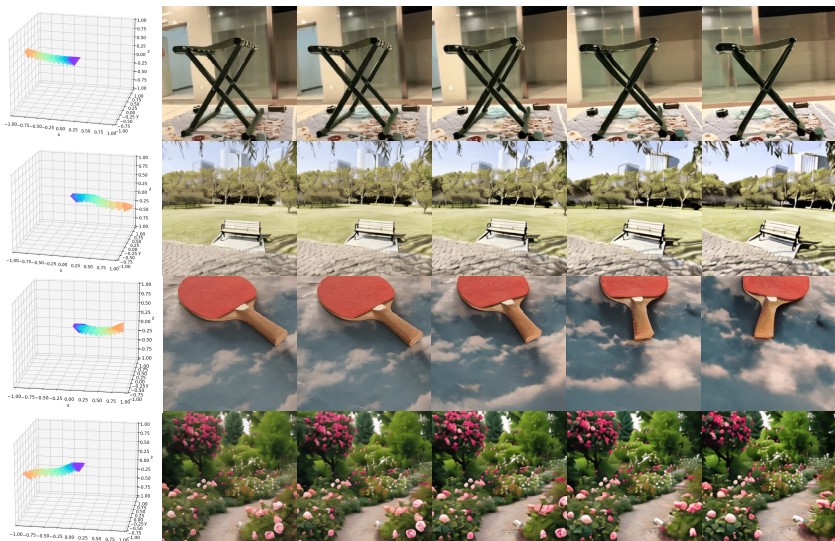

Figure 1: Given a single frame (the first image column) and a sequence of cameras as input, our CamCo model is able to synthesize videos that follow the camera conditions with 3D consistency. We support indoor, outdoor, object-centric, and text-to-image generated images. The prompt for the last row is "A lush garden filled with blooming roses of various colors, with a gravel path winding through it". The camera of the first frame starts from world origin, shown in purple.

## 1 Introduction

The rapid evolution of diffusion-based generative models (Rombach et al., 2022; Saharia et al., 2022; Ramesh et al., 2021; 2022; Betker et al., 2023) has empowered general users to generate video content from textual or visual inputs easily. Recent advancements in video generative models (Khachatryan

et al., 2023; Blattmann et al., 2023; AI, 2023; Guo et al., 2023b) have underscored the importance of user control (Wu et al., 2023; Guo et al., 2023b;a; Peruzzo et al., 2024) over the generated content. Rather than relying on the cumbersome and error-prone process of prompt engineering, controllable generation techniques ensure that the output adheres to more specified and fine-grained control signals, thereby enhancing user satisfaction by producing more desirable outcomes. Recent studies (Zhang & Agrawala, 2023; Li et al., 2023; Mou et al., 2024) have incorporated additional training layers to integrate these control signals, further refining content generation capabilities.

Despite the diverse control signals available (e.g., depth, edge map, human pose) (Wu et al., 2023; Guo et al., 2023b;a; Peruzzo et al., 2024; Zhang & Agrawala, 2023; Li et al., 2023; Mou et al., 2024), controlling camera viewpoints in generated content has received little attention. Camera motion, a crucial filmmaking technique (Nielsen et al., 2007), enables content creators to shift the audience's perspective without cutting the scene (Sikov, 2020), thereby conveying emotional states effectively (Yilmaz et al., 2023). This technique is vital for ensuring that videos are practically usable in downstream applications such as augmented reality, filmmaking, and game development (Heimann et al., 2014). It allows creators to communicate more dynamically with their audience and adhere to a pre-designed script or storyboard.

To enable camera control in content generation, early methods (Guo et al., 2023b;a) utilize a fixed number of categories to represent camera trajectories, providing a coarse control over the camera motion. They use LoRA (Hu et al., 2021) to fuse the camera information and conditionally generate videos that belong to certain categories of camera motion. To improve the camera motion granularity, (Wang et al., 2023b) has proposed to use adapter layers to accept normalized camera extrinsic matrices. However, this approach uses one-dimensional numeric values to represent the camera pose, which struggles to precisely control the video content when provided with complex camera motion.

To overcome the above issues, we introduce **CamCo**, a **cam**era-**co**ntrollable image-to-video generation framework that produces 3D-consistent videos. Our framework is built upon a pre-trained image-to-video diffusion model, preserving the majority of the original model parameters to maintain its generative capabilities. To enhance the accuracy of camera motion control, we represent camera pose with Plücker coordinates that encode both camera intrinsics and extrinsics into a pixel-wise embedding, which is a dense conditioning signal for video generation. Moreover, to improve the geometric consistency of synthesized videos, we introduce a new epipolar constraint attention module in each attention block to enforce epipolar constraints across frames. Finally, we implement a data curation pipeline that annotates in-the-wild video frames with estimated camera poses, enhancing our capability of generating videos with large object motions. Our experiments on various domains demonstrate that **CamCo** outperforms previous state-of-the-art methods in terms of visual quality, camera controllability, and geometry consistency.

Our contributions can be summarized as follows,

- We propose **CamCo**, a novel **cam**era-**co**ntrollable image-to-video generation framework that can generate high-quality, 3D-consistent videos.
- We build the first 3D-consistent video diffusion model by adapting a pre-trained image-to-video diffusion model. We parameterize camera information via Plücker coordinates and incorporate epipolar constraints via new epipolar constraint attention modules.
- We introduce a data curation pipeline to handle in-the-wild videos with dynamic subjects and fine-tune CamCo on the curated dataset to enhance its ability to generate videos with both camera ego-motion and dynamic subjects.
- Our method exhibits superior 3D consistency, visual quality, and camera controllability when compared with previous works.

## 2 RELATED WORK

### 2.1 DIFFUSION-BASED VIDEO GENERATION

The recent advances in diffusion models (Rombach et al., 2022; Saharia et al., 2022; Ramesh et al., 2021; 2022; Betker et al., 2023) have provided users with enhanced flexibility in visualizing their imaginations. Leveraging large-scale image-text paired datasets (Schuhmann et al., 2022; Byeon

et al., 2022), diffusion models have become the state-of-the-art generators for text-to-image (T2I) synthesis. However, due to the lack of high-quality video-text datasets (AI, 2023; Blattmann et al., 2023), researchers have extensively explored adapting text-to-image generators into text-to-video (T2V) generators. Text2Video-Zero (Khachatryan et al., 2023) builds the first training-free video generation pipeline. AnimateDiff (Guo et al., 2023b) constructs a motion module applicable to various base T2I models. VideoLDM (Blattmann et al., 2023) and SVD (AI, 2023) inflate the T2I models with additional temporal layers and train on curated datasets to improve the visual quality. The recently released Sora model (Brooks et al., 2024) has demonstrated impressive video generation results, drawing attention to transformer-based diffusion backbones (Peebles & Xie, 2023; Ma et al., 2024; Yu et al., 2023; 2024). Our work builds upon the open-source SVD (AI, 2023) model. We introduce novel camera conditioning and geometry-aware blocks to achieve camera controllability and geometry consistency for the challenging image-to-video generation task.

## 2.2 CONTROLLABLE CONTENT CREATION

Along with the rapid development of generative models that produce content from various input modalities, improving user control over generation has also attracted significant attention. ControlNet (Zhang & Agrawala, 2023), T2I-adapter (Mou et al., 2024), and GLIGEN (Li et al., 2023) pioneered the introduction of control signals, including depth, edge maps, semantic maps, and object bounding boxes, to text-to-image generation. These approaches ensure the synthesis backbone (Rombach et al., 2022) remains intact while adding trainable modules to maintain both control and generative capabilities. Similar approaches have been applied to video generation, supporting controls like depth, bounding boxes, and semantic maps (Guo et al., 2023b;a; Wu et al., 2023; Peruzzo et al., 2024). However, controlling camera motion has received limited attention. AnimateDiff (Guo et al., 2023b) and SVD (AI, 2023) explore class-conditioned video generation, clustering camera movements, and using LoRA (Hu et al., 2021) modules to generate specific camera motions. MotionCtrl (Wang et al., 2023b) enhances control by employing camera extrinsic matrices as conditioning signals. While effective for simple trajectories, their reliance on 1D numeric values results in imprecise control in complex real-world scenarios. Our work significantly improves granularity and accuracy in camera control by introducing Plücker coordinates and geometry-aware layers to image-to-video generation. Concurrently, CameraCtrl (He et al., 2024) also employs Plücker coordinates but focuses on the text-to-video generation while paying less attention to 3D consistency.

## 2.3 3D SCENE SYNTHESIS

Due to the lack of high-quality 3D scene datasets (Deitke et al., 2023b;a), it is challenging to adapt the success of image and video generation (Betker et al., 2023; Ramesh et al., 2021; Rombach et al., 2022; Saharia et al., 2022; Yu et al., 2023; 2024) into 3D generation tasks (Liu et al., 2023; Shi et al., 2023). DreamFusion (Poole et al., 2022) addresses this issue by introducing score distillation sampling, which distills knowledge from 2D diffusion models. This technique has been successfully applied to 3D object synthesis from text (Lin et al., 2023; Tang et al., 2023; Wang et al., 2023a) and image (Xu et al., 2022; Liu et al., 2023; Shi et al., 2023) input. However, due to the limited ability of 2D diffusion models to generate complex scenes, score distillation is not directly applicable to scene generation. Consequently, compositional generation (Po & Wetzstein, 2023; Zhang et al., 2023; Xu et al., 2024; Gao et al., 2023) has emerged as a popular approach to tackle these challenges. Video generators (Voleti et al., 2024; Melas-Kyriazi et al., 2024; Zuo et al., 2024) have also been adapted to generate simple objects by conditioning the three degrees of freedom (DoF) camera information on the time embeddings of diffusion models. However, these approaches are not directly applicable to real-world scenarios because their simplifications of camera information do not suffice for real videos that contain diverse extrinsic and intrinsic parameters. Our work focuses on generating 3D-consistent videos from in-the-wild images, providing insights into the challenging direction of 3D scene synthesis.

## 3 METHOD

In this section, we present our novel method for generating camera-controllable, geometry-consistent videos, as shown in Fig. 2. First, we outline the preliminaries of the pre-trained image-to-video diffusion model. Then, we introduce our camera parameterization and control modules, which

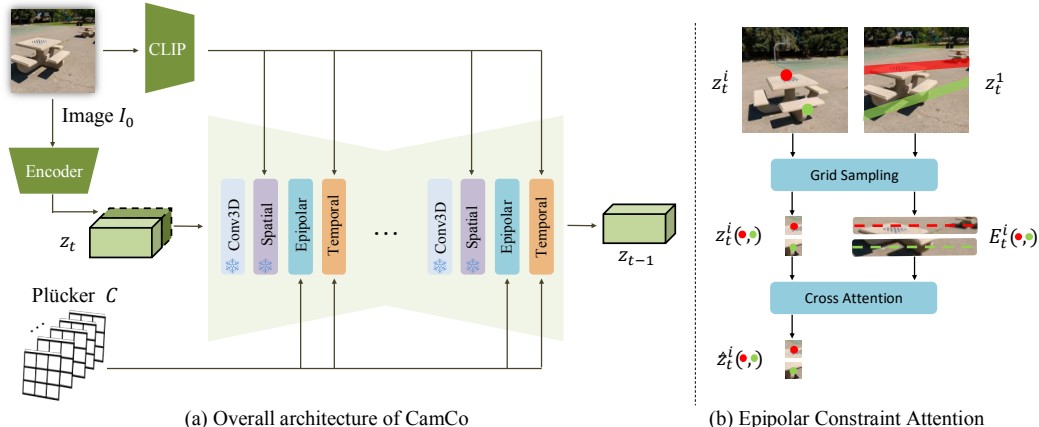

(a) Overall architecture of CamCo         (b) Epipolar Constraint Attention

Figure 2: Overview of our proposed CamCo framework. (a) shows the architecture, where we introduce Plücker coordinates as an effective camera parameterization and an epipolar constraint attention block to enforce geometry consistency. (b) illustrates our epipolar constraint attention block. For each queried pixel via grid sampling from the $i$-th frame $z_t^i$, we gather information from the corresponding epipolar line in the source frame $z_t^1$ using a cross-attention layer. The features $E_t^i$ along the latent space epipolar line encode the local regions around it in the image space.

integrate camera control into the pre-trained video diffusion model. Further, we describe our epipolar constraint attention mechanism, ensuring the geometric consistency of generated videos. Finally, we discuss how to curate data annotations to generate object motion effectively.

### 3.1 IMAGE-TO-VIDEO GENERATION

**Problem Setting** The task of image-to-video generation involves using a single image $I_0$ as input of a video generator to produce a sequence of output frames $O_1, \cdots, O_n$ with temporal consistency. In our experiments, $n$ is typically set to 14. To add camera control, we retain the original input and output format while additionally introducing a set of camera information $C_1, \cdots, C_n$. These camera parameters $C_i$ are used as fine-grained conditioning of the video generator, ensuring that the generated frames follow the viewpoint changes specified by the camera sequence.

**Base Model Architecture** We train our CamCo based on a publicly available pre-trained image-to-video diffusion model, Stable Video Diffusion (SVD) (AI, 2023). SVD is built upon Stable Diffusion 2.1 model (Rombach et al., 2022) that is originally trained through an EDM (Karras et al., 2022) framework. SVD inserts temporal convolution and attention layers after spatial layers following VideoLDM (Blattmann et al., 2023). Unlike previous works that only train the temporal layers (Blattmann et al., 2023; Guo et al., 2023b) or are training-free (Khachatryan et al., 2023), SVD fine-tunes all parameters.

**Training** SVD incorporates a continuous-time noise scheduler (Karras et al., 2022), which supports a continuous range of noise levels. The diffusion model learns to gradually denoise a high-variance Gaussian noise towards the (video) data distribution $\mathbf{x}_0 \sim p_0$. Let $p(\mathbf{x}; \sigma(t))$ denote the marginal probability of noisy data $\mathbf{x}_t = \mathbf{x}_0 + \mathbf{n}(t)$ where the added Gaussian noise $\mathbf{n}(t) \sim \mathcal{N}(0, \sigma^2(t)\mathbf{I})$, the iterative denoising process corresponds to the probability flow ordinary differential equation (ODE):

$$d\mathbf{x} = -\dot{\sigma}(t)\sigma(t)\nabla_\mathbf{x} \log p(\mathbf{x}; \sigma(t))dt, \tag{1}$$

where $\nabla_\mathbf{x} \log p(\mathbf{x}; \sigma(t))$ is the score function parameterized by a neural network $D_{\boldsymbol{\theta}}$ through $\nabla_\mathbf{x} \log p(\mathbf{x}; \sigma) \approx (D_{\boldsymbol{\theta}}(\mathbf{x}; \sigma) - \mathbf{x})/\sigma^2$. The training objective is denoising score matching:

$$\mathbb{E}\left[\|D_{\boldsymbol{\theta}}(\mathbf{x}_0 + \mathbf{n}; \sigma, \mathbf{c}) - \mathbf{x}_0\|_2^2\right]. \tag{2}$$

where $\mathbf{c}$ denotes the conditioning information (e.g., the input image $I_0$).

Classifier-free guidance (Ho & Salimans, 2022) (CFG) is also adopted for better conditioning following. During training, the condition signal $\mathbf{c}$ is randomly set to a zero tensor $\emptyset$ to properly learn an unconditional model $D(\mathbf{x}; \emptyset)$ with a 10% probability. At inference time, CFG is formulated as follows,

$$D_\omega(\mathbf{x}; \mathbf{c}) = \omega(D(\mathbf{x}; \mathbf{c}) - D(\mathbf{x}; \emptyset)) + D(\mathbf{x}; \emptyset), \tag{3}$$

where $\omega$ is the weighting coefficient. Following SVD (AI, 2023), $\omega$ is empirically set to linearly increase from 1 for the first frame and 3 for the last frame.

## 3.2 Injecting Camera Control to Video Diffusion Model

**Camera Parameterization** Unlike generating canonicalized 3D objects that operate in a 3-DoF camera space, video diffusion models dealing with real-world dynamic scenes must manage 6-DoF cameras with diverse intrinsic parameters. Consequently, representing camera information using elevation (Voleti et al., 2024) or camera extrinsics (Wang et al., 2023b) is suboptimal. Moreover, using one-dimensional numeric values for camera extrinsic as conditioning leads to imprecise camera control (Wang et al., 2023b) in challenging camera trajectories. To comprehensively represent both camera extrinsic and intrinsic information, we draw inspiration from light field networks (Sitzmann et al., 2021) and adopt Plücker coordinate (Jia, 2020). Specifically, let $o$ be the ray origin and $d$ be the ray direction of a pixel. The Plücker coordinate is formulated as $P = (o \times d', d')$, where $\times$ represents the cross product and $d'$ is the normalized ray direction $d' = \frac{d}{||d||}$. Given camera extrinsics $E = [\mathbf{R}|\mathbf{T}]$ and intrinsics $\mathbf{K}$, the ray direction $d_{u,v}$ for 2D pixel located at $(u, v)$ is defined as $d = \mathbf{R}\mathbf{K}^{-1}(\begin{smallmatrix} u \\ v \\ 1 \end{smallmatrix}) + \mathbf{T}$. All camera poses are defined relatively to the first frame. Plücker coordinates uniformly represent rays, making them a suitable positional embedding for the network, and they have proven successful in parameterizing 360-degree unbounded light fields (Sitzmann et al., 2021).

**Camera Control Module** To incorporate the camera embeddings, we add a simple adapter layer to each temporal attention block in the pre-trained video generator. This design best preserves the generation abilities the base model acquired during pre-training (Zhang & Agrawala, 2023; Mou et al., 2024; Li et al., 2023). Specifically, in each temporal attention block, we first concatenate the Plücker coordinates with the network features along the channel dimension, and then pass them into the $1 \times 1$ convolution layer to project back to the original feature space. The projected features are passed to the remaining temporal attention layers. Plücker embeddings are downsampled via interpolation if their spatial resolution mismatches with the network features at bottleneck layers. Inspired by ControlNet (Zhang & Agrawala, 2023), we zero-initialize partial weights of the convolution layer so that at the start of training, the whole network remains unaffected by these additional layers, and we gradually add the camera control through the gradient descent.

## 3.3 Ensuring 3D-Consistent Generation

**Drawback of vanilla architecture** While using the Plücker coordinates results in more fine-grained camera control, it does not directly ensure the geometric consistency of the generated videos. This inconsistency issue is primarily rooted in the inefficiency of the vanilla architecture of video diffusion models (e.g., SVD) in modeling geometric relationships across frames. Recall that the vanilla architecture introduces dense self-attention in spatial and temporal dimensions, where any pixel at any frame is free to attend to any other pixels, potentially leading to pixel-copying behaviors (Kant et al., 2024) that appear correct but do not adhere to geometric constraints. To address this issue, we need to design an attention masking mechanism to ensure each pixel attends to features that are geometrically correlated.

**Epipolar Constraint Attention (ECA)** Due to the missing depth information from the in-the-wild input images, we do not have the exact pointwise correspondences across frames. Nevertheless, for each pixel in the target view, we can derive an epipolar line in the source view. Epipolar constraint describes that the corresponding point of a feature in one image must lie on the epipolar line associated with that feature's projection in another image. We then propose epipolar constraint attention (ECA), which conducts cross-attention between the features on the epipolar line and features at the target location, ensuring that the current frames adhere to projective geometry. We visualize our ECA block in Fig. 2(b) and provide more details in Sec. A.

Our epipolar constraint attention associates the source view $O_1$ with the target views $O_i, i \in [2, n]$ through the epipolar lines. Due to the diverse camera pose sequences, the epipolar lines can vary in direction and length, posing a challenge for constructing epipolar lines that efficiently support batch attention calculation. To ensure efficiency, we propose constructing the epipolar lines by resampling the visible pixel locations, thereby avoiding variable-length epipolar line features.

At timestep $t$, we represent $z_t^i \in \mathrm{R}^{hw \times d}$ as the latent feature of $i$-th frame and $E_t^i \in \mathrm{R}^{hw \times l \times d}$ as the point features along the corresponding epipolar lines, sampled from the feature map of the first frame, where $l$ is the number of points on the epipolar line. Denote the query, key, and value by $q := z_t^1 W_q \in \mathbb{R}^{hw \times 1 \times d}$, $k := E_t^i W_k \in \mathbb{R}^{hw \times l \times d}$, and $v := E_t^i W_v \in \mathbb{R}^{hw \times l \times d}$, respectively, where $W_q$, $W_k$ and $W_v$ are the projection matrices. Then, ECA between frame $i$ and the first frame is given by $\mathrm{ECA}(z_t^i, E_t^i) = \sigma(\frac{qk^T}{\sqrt{d}})v \in \mathbb{R}^{hw \times d}$. Unlike previous works that apply spatial binary masks (Kant et al., 2024) or weight maps (Tseng et al., 2023) to the original cross-attention in $O((hw)^2)$, our attention mechanism implements the cross-attention matrix of size $O(hwl)$. This efficient design avoids unnecessary computation and improves the efficiency of imposing the epipolar constraint by one order of magnitude (assuming $l \approx O(h) \approx O(w)$), which is crucial for video generation where multiple frames are generated simultaneously. For training, we only update the weights in the ECA layers while freezing the base video model. This preserves the base model's generation and generalization abilities.

### 3.4 Improving Object Motion

While the above design enforces good camera control and geometric consistency, the model can easily overfit to generate only static scenes with little object motion because it lacks the training data with dynamic scenes.

**Augmented Training Dataset with better Motion**  To overcome this issue, we need to augment our training dataset with dynamic videos that contain rich object motion. However, the number of available dynamic videos with camera poses is greatly limited due to the difficulties in collecting annotations. Consequently, we first randomly sample videos from the WebVid (Bain et al., 2021) dataset, which only contains video frames and lacks camera annotations. Then, we annotate the camera poses in each video clip using Particle-SfM (Zhao et al., 2022). Particle-SfM is a pre-trained framework capable of estimating camera trajectories for monocular videos captured in dynamic scenes. Due to the time-consuming nature of this process, we randomly sample 32 frames from each original video to feed into the Particle-SfM (Zhao et al., 2022) pipeline.

**Curating High-quality Samples**  Note that the WebVid dataset contains videos of various kinds, including many with static cameras. To ensure the model learns to handle more challenging motions, we need to filter out sequences where the estimated camera trajectories show minimal displacement. Due to the inevitable ambiguity between object motion and camera motion, in-the-wild estimation of camera poses can be inaccurate. We thus use the number of points in the reconstructed sparse point clouds from Particle-SfM (Zhao et al., 2022) as an indicator of camera annotation quality. The intuition is that an accurate camera pose estimation comes from well-registered frames. If the frames have more 3D-consistent pixels, more point correspondences will contribute to solving the camera parameters. With this indicator, we only use videos with accurate camera pose estimation and high object motion, and construct a training set of high-quality 12,000 video sequences annotated with camera poses.

## 4 Experiments

### 4.1 Implementation Details

Our CamCo model builds upon SVD (AI, 2023), which is a UNet-based open-source image-to-video diffusion model with architecture similar to VideoLDM (Blattmann et al., 2023). In our experiments, we use a relative camera system where all camera poses are converted to become relative to the first frame. The camera of the first frame is placed to the world origin and rotated to face towards the x-axis. During training, we randomly sample the strides of the image sequences to augment the speed of the video. The sampled camera extrinsics are then normalized to have a unit maximum distance

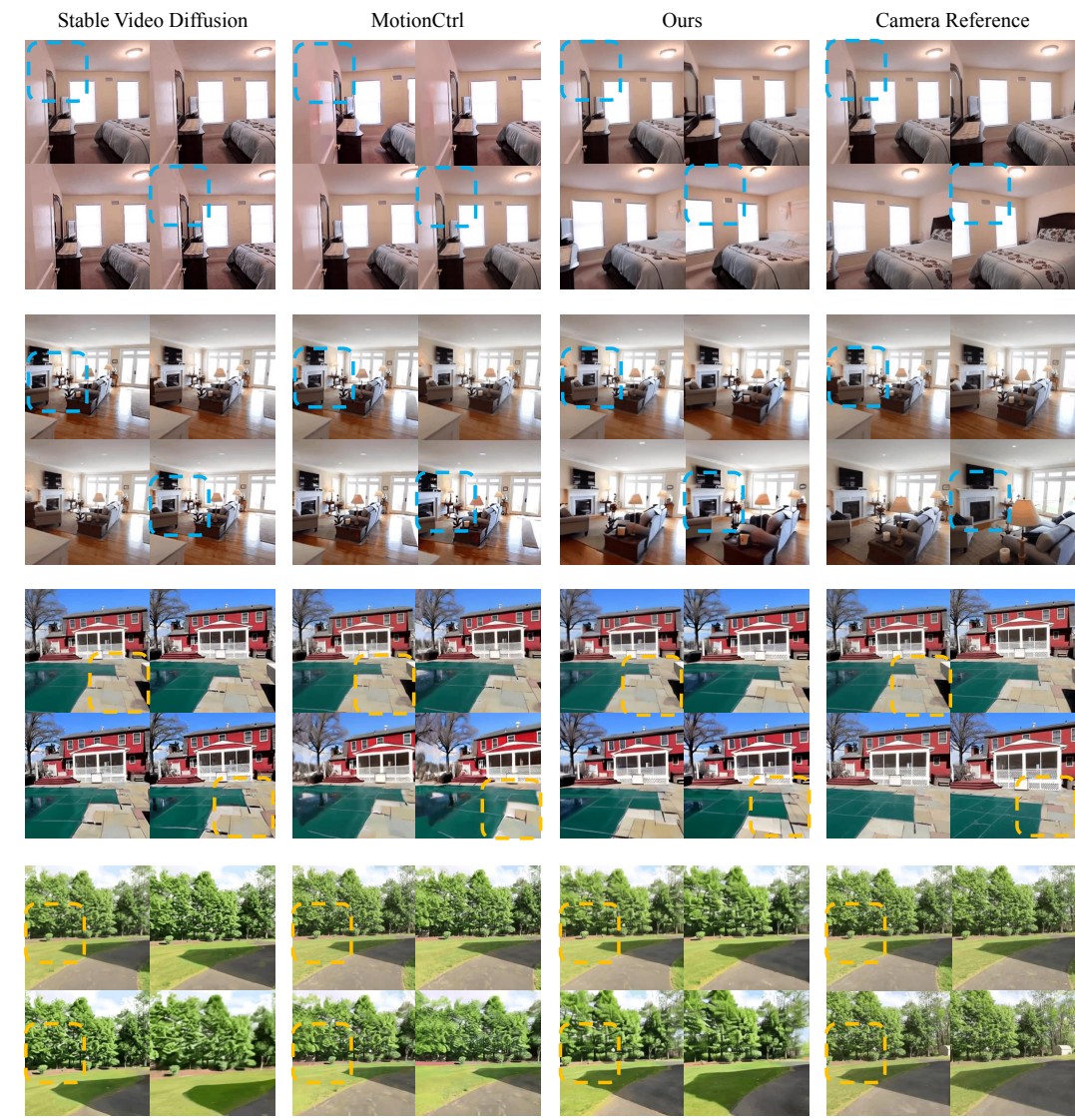

| Stable Video Diffusion | MotionCtrl | Ours | Camera Reference |

Figure 3: Static scene video generation results. The last column provides reference videos that visualize the camera trajectories. The images and trajectories are unseen during training. Regions are highlighted to reveal camera motion. Please check the video results for better visualizations on the project page: `https://camco2024.github.io/`.

against the world origin to stabilize the training. For simplicity and efficiency, the training frames are first center-cropped to square images and then downsampled to $256 \times 256$ resolution. During inference, we use 25 sampling steps to obtain the 14 frames. The decoding chunk size of the latent decoder is set to 14 frames for the best visual quality. Our model is trained with batch-size 64 in total on 16 A100 GPUs for around two days. The learning rate is set to 2e-5 with a warmup of 1,000 steps using the Adam optimizer. We use fp16 training implemented with "accelerate"[1] library.

## 4.2 BASELINES

We mainly compare our method against Stable Video Diffusion (AI, 2023), VideoCrafter (Chen et al., 2023), and MotionCtrl (Wang et al., 2023b). For MotionCtrl (Wang et al., 2023b), we take the image-to-video generation checkpoint released by the authors for a fair comparison since their

---

[1] `https://github.com/huggingface/accelerate`

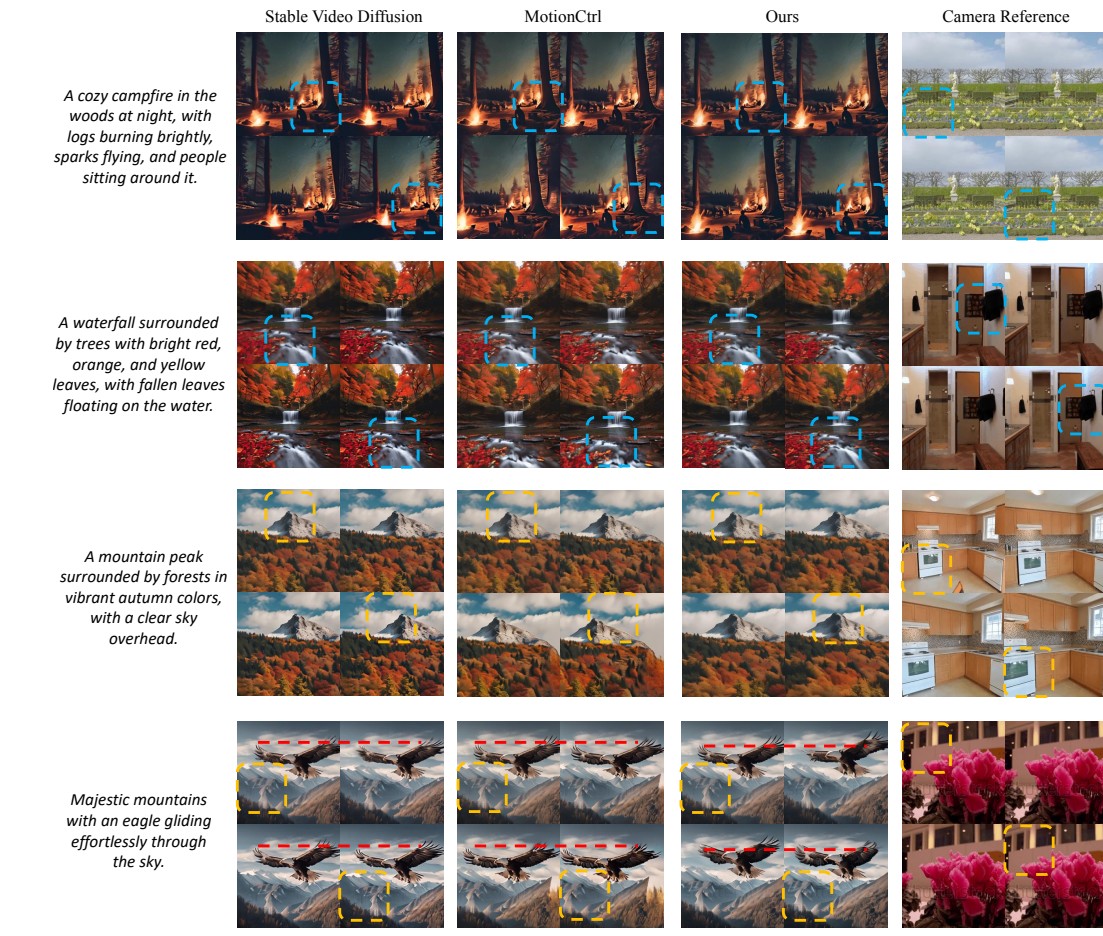

Figure 4: Dynamic scene video generation results where the first frame is generated by SDXL (Podell et al., 2023) from the prompt in the left. The last column provides reference videos that visualize the camera trajectories. The trajectories are unseen during training. Regions are highlighted to reveal camera motion and object motion better. Please check the video results for better visualizations on the project page: https://camco2024.github.io/.

released checkpoint is also trained from Stable Video Diffusion (AI, 2023). For each sequence in RealEstate10k (Zhou et al., 2018), we first randomly select a starting frame and then take every 8th frame to ensure reasonable camera change. Note that in our evaluations and visualizations, none of the input images or camera trajectories were seen during training.

### 4.3 QUANTITATIVE COMPARISONS

Since our major goal is to generate videos that follow the camera motion input, we need to measure the camera pose from the output videos. Thanks to COLMAP (Schönberger & Frahm, 2016; Schönberger et al., 2016), we are able to annotate the camera poses of videos we obtain. Because structure-from-motion algorithms can only estimate the scene structure up to a certain scale, we further canonicalize the estimated camera poses before calculating the differences. Specifically, we follow our training protocol where we first convert the camera systems to be relative to the first frame and then normalize the scale of the scene by finding the furthest camera against the first frame.

**COLMAP error rate and matched points** Due to the randomness introduced in COLMAP's pipeline, we report the average rate with five trials for each video. We consider a reconstruction to be successful only when a sparse model is created with all 14 frames matched together. Similar to

Table 1: Quantitative comparison against baseline methods on static videos. * denotes that the results of these metrics are averaged for sequences that are successfully processed by COLMAP.

| Method | FID ↓ | FVD ↓ | COLMAP error↓ | Points*↑ | Translation* ↓ | Rotation* ↓ |
|---|---|---|---|---|---|---|
| VideoCrafter | 30.58 | 342.66 | 93.9% | 263.12 | 4.2882 | 9.2891 |
| Stable Video Diffusion | 18.59 | 281.45 | 64.3% | 434.18 | 2.8476 | 9.5735 |
| MotionCtrl | 14.74 | 229.33 | 14.6% | 447.93 | 3.0445 | 8.9289 |
| Ours | **14.66** | **138.01** | **3.8%** | **461.07** | **2.6655** | **7.0218** |

Table 2: Ablation studies on model variants. * denotes that the results of these metrics are averaged for sequences that are successfully processed by COLMAP.

| Method | FID ↓ | FVD ↓ | COLMAP error↓ | Points*↑ | Translation* ↓ | Rotation* ↓ |
|---|---|---|---|---|---|---|
| Time Embedding | 16.08 | 152.81 | 12.9% | 453.46 | 3.0327 | 7.7283 |
| 1-Dim Camera | 16.12 | 150.57 | 12.7% | 452.98 | 3.1272 | 7.8846 |
| w/o Plücker | 15.95 | 144.16 | 14.7% | 458.42 | 2.7335 | 7.2427 |
| w/o Epipolar Attention | 15.51 | 144.30 | 10.2% | 458.71 | 3.0502 | 7.2215 |
| Full Model | **14.66** | **138.01** | **3.8%** | **461.07** | **2.6655** | **7.0218** |

CO3D (Reizenstein et al., 2021), we further report the number of points available in the reconstructed sparse point clouds as a reflection of the 3D consistency of the video frames.

**Pose Accuracy**   We extract the camera-to-world matrices of the COLMAP predictions and obtain the estimated rotation and translation vectors, similar to CameraCtrl (He et al., 2024). The relative rotation distances are then converted to radians, and we sum the total error of 14 frames,

$$R_{\text{err}} = \sum_{i=1}^{n} \arccos(\frac{\text{tr}(R_{\text{out}_i}^T R_{\text{gt}_i}) - 1}{2}). \qquad (4)$$

After normalizing the maximum distance-to-origin to 1, the norm of the relative translation vector for each frame is also summed together to form the translation error of the whole video,

$$T_{\text{err}} = \sum_{i=1}^{n} \left\| T_{\text{out}_i} - T_{\text{gt}_i} \right\|_2. \qquad (5)$$

**FID and FVD**   Additionally, we evaluate the visual quality of the generated video frames, regardless of the camera following ability or geometry quality. FID (Heusel et al., 2017) and FVD (Unterthiner et al., 2018) are calculated based on deep features extracted from video frames. We, therefore, use FID (Heusel et al., 2017) and FVD (Unterthiner et al., 2018) to measure the distance between the generated frames and the corresponding reference videos.

### 4.4 QUALITATIVE COMPARISONS

**Static Generation**   Due to the lack of large-scale camera pose annotation of dynamic videos in the wild, we quantitatively evaluate our method against previous works on static videos. Specifically, we randomly sampled 1,000 videos from the Realestate-10k (Zhou et al., 2018) test set. The camera poses were originally annotated using COLMAP (Schönberger & Frahm, 2016; Schönberger et al., 2016). We provide qualitative and quantitative comparisons in Fig. 3 and Tab. 1, respectively. In Fig. 3, the ground truth cameras are visualized through the camera reference videos. Compared to the baselines, our method produces results with the best camera-following ability. In contrast, the results produced by Stable Video Diffusion do not follow the camera control, and MotionCtrl demonstrates inaccurate camera-following results. Our results outperform the baselines by a large margin in all metrics, indicating our superiority in visual quality, camera controllability, and geometric consistency. Note that for all experiments, neither the input image nor the camera trajectory was seen during training.

**Dynamic Generation**   Similar to the static scenario, we evaluate our method against previous works on randomly sampled 1,000 videos from the test split of our annotated WebVid (Bain et al., 2021) dataset. We provide qualitative and quantitative comparisons in Fig. 4 and Tab. 3, respectively.

Table 3: Quantitative comparison on generated dynamic videos.

| Method | SVD | MotionCtrl | Static Only | Uncurated Dynamic | Full Model |
|---|---|---|---|---|---|
| FID ↓ | 27.43 | 36.21 | 30.78 | 23.22 | **22.19** |
| FVD ↓ | 121.05 | 188.86 | 169.48 | 148.08 | **137.59** |

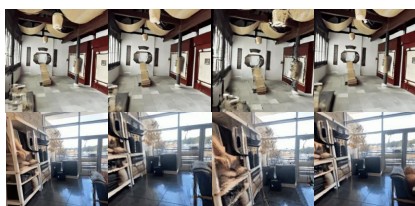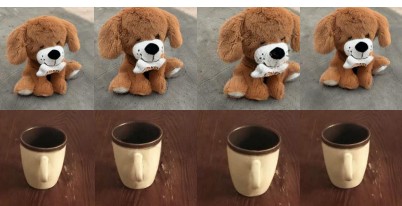

Figure 5: Rendered novel views from the 3D reconstruction results of frames generated by CamCo.

In Fig. 4, we provide the generated videos when inferencing text-to-image results produced by SDXL (Podell et al., 2023). The last column provides the camera trajectories. Compared to the baselines, our method excels at producing perspective changes as well as object motion. Notably, in the eagle case, the baselines fail to produce reasonable object motion or follow the camera change configurations, whereas our method successfully generates both camera changes and object movements vividly. In Tab. 4, we provide FID and FVD calculated against ground truth videos sampled from the WebVid dataset. Our full model produces the best FID and FVD values, indicating the model produces the best visual quality. In comparison, the model trained on the uncurated dynamic dataset and the static version model show inferior results, indicating poorer object motion. Note that for all experiments, neither the input image nor the camera trajectory was seen during training.

### 4.5 ABLATION STUDIES

In this section, we conduct several ablation studies on the model variants, including different versions of camera conditioning and the importance of our epipolar attention block. "w/o epipolar attention" denotes the baseline where we remove the epipolar constraint attention from our model and only use the temporal convolution layers. "w/o Plücker" refers to the model variant where we remove the Plücker coordinate input in the temporal blocks and rely on the epipolar constraint attention to modulate the camera control. "1-dim camera" means using 1-dimensional camera matrices as the conditioning signal for modulating the features in temporal blocks instead of using 2-dimensional Plücker coordinates. "Time embedding" refers to modulating the time embedding feature with camera matrices instead of using Plücker coordinates. As can be seen in Tab. 2, the full model delivers the best results in both 2D visual quality (FID and FVD) and camera pose accuracy (COLMAP error, points, translation, and rotation error). In contrast, the model variants lack the ability of accurately controlling the camera poses which can lead to deteriorated visual appearance.

### 4.6 3D RECONSTRUCTION OF GENERATED FRAMES

In Fig. 5, we provide novel view renderings of the 3D reconstruction results for our generated frames. These novel view renderings demonstrate that the 3D reconstruction has obtained sharp geometry without floaters, indicating the outstanding geometric consistency of the output frames of CamCo. These results are hard to obtain using previous methods. Please check the video results for better visualizations on the project page: `https://camco2024.github.io/`.

## 5 CONCLUSION

In this paper, we introduced CamCo, a camera-controllable framework that produces 3D-consistent videos. Our method is built upon a pre-trained image-to-video diffusion model and introduces novel camera conditioning and geometry constraint blocks to ensure camera control accuracy and geometry consistency. Further, we proposed a data curation pipeline to improve the generation of object motion. Our experiments on images of various domains demonstrate the superiority of CamCo against previous works in terms of camera controllability, geometry consistency, and visual quality.

## 6 REPRODUCIBILITY STATEMENT

We provide sufficient details for reproducing our method in the main paper and the appendices. We also specify training hyperparameters, such as learning rate scheduling, batch size, etc. For evaluation, we provide detailed setup for configuring baselines and COLMAP for best reproduction. For RealEstate-10k (Zhou et al., 2018), we use the train/test split released by the authors of Pixel-Splat (Charatan et al., 2023). At the time of working on this project, the WebVid dataset was available for research purposes, and we do not intend to use the models trained in this paper for commercial purposes.

## 7 ETHICS STATEMENT

The paper focuses on generating videos from image input. Like all other large models trained from large-scale datasets, our model can contain certain social biases. Such biases could perpetuate stereotypes or misrepresentations in generated videos, thus influencing public perceptions and potentially leading to societal harm. Therefore, generative video models generally need to be applied with caution.

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

## A  ADDITIONAL DETAILS ON EPIPOLAR CONSTRAINT ATTENTION

An epipolar line refers to the projection on one camera's image plane of the line connecting a point in 3D space with the center of projection of another camera. Specifically, let $\mathbf{K}, \mathbf{R}, \mathbf{T}$ be the relative camera pose between frame $i$ and the first frame, the projections of point $p$ and camera origin $o = \left(\begin{smallmatrix} 0 \\ 0 \\ 0 \end{smallmatrix}\right)$ on source view the first frame are denoted as $\mathrm{Proj}\left(\mathbf{R}\left(\mathbf{K}^{-1}\mathbf{p}^i\right) + \mathbf{T}\right)$ and $\mathrm{Proj}\left(\mathbf{R}\left(\mathbf{K}^{-1}\left(\begin{smallmatrix} 0 \\ 0 \\ 0 \end{smallmatrix}\right)\right) + \mathbf{t}\right)$, respectively, where Proj is the projection function. The epipolar line is, therefore, given by

$$\mathbf{L} = \mathbf{o} + c\left(\mathbf{p} - \mathbf{o}\right), \tag{6}$$

where $c \in \{0, \infty\} \in \mathbb{R}$.

## B  EXPERIMENT DETAILS

### B.1  COLMAP CONFIGURATION

We use the widely adopted COLMAP configuration for few-view 3D reconstruction (Deng et al., 2022) and assume all frames in each video share the same camera intrinsics. For the feature extractor, we set -SiftExtraction.estimate_affine_shape 1 -SiftExtraction.domain_size_pooling 1 -ImageReader.single_camera 1 . For the feature matching process, we set -SiftMatching.guided_matching 1 -SiftMatching.max_num_matches 65536. For each video, we retry the process five times, at most.

### B.2  PARTICLE-SFM WORKFLOW

We use the full default configuration of Particle-sfm for the best performance. Videos are processed at their original resolution without any resizing or cropping operation. Given a long video sequence, we randomly sample a frame stride and then uniformly obtain a set of 32 frames according to the stride. The 32 frames are then sent into Particle-sfm for camera annotation.

### B.3  ADDITIONAL SETTINGS

We use the same train-test split of RealEstate-10k (Zhou et al., 2018) as in PixelSplat (Charatan et al., 2023). Baseline methods are ran at their originally designed resolution. Thanks to the image-to-video nature, when the input image is at the resolution of $256 \times 256$, despite what the output image resolution is, the output becomes undistorted when being resized to $256 \times 256$. For baselines, we use the open-source checkpoints released by the authors. We use "clean-fid"[2] and "common-metrics-on-video-quality"[3] for calculating FID and FVD, respectively. For FVD, we report the results in VideoGPT (Yan et al., 2021) format.

## C  ADDITIONAL QUALITATIVE RESULTS

We have prepared additional qualitative results in our supplementary, such as comparisons with baselines including concurrent work CameraCtrl, 3D reconstruction results from CamCo's generated videos, CamCo evaluated on in-the-wild images, and qualitative comparisons for ablation studies. Please check the video results for better visualizations on the project page: https://camco2024.github.io/. Note that in our evaluations and visualizations, none of the input images or camera trajectories were seen during training.

## D  LIMITATIONS AND FUTURE WORK

Because our training data are video frames with the same camera intrinsics, our model generates images that have the same camera intrinsic as the input image. Therefore, the model can not generate

---

[2]https://github.com/GaParmar/clean-fid
[3]https://github.com/JunyaoHu/common_metrics_on_video_quality

complex camera intrinsic changes such as dolly zoom. Additionally, CamCo is able to generate 14 frames at the resolution of $256 \times 256$, which only covers limited viewpoints and can be insufficient for large scenes. We will explore generating longer, and larger resolution videos in future work.

