# OpenReview forum: "CamCo: Camera-Controllable 3D-Consistent Image-to-Video Generation"
_ICLR.cc/2025/Conference — Submitted to ICLR 2025_

### Official Review · Reviewer_XEZi · 2024-10-29

**Soundness:** 3
**Presentation:** 4
**Contribution:** 1
**Rating:** 3
**Confidence:** 4

**Summary:**

The paper introduces a mechanism that adds camera control to a text-to-video diffusion model, i.e., SVD. For this purpose, the authors propose to finetune the model on the WebVid dataset using a conditioning mechanism that uses Plucker coordinates and epipolar attention to improve consistency. The proposed method is shown to outperform the MotionCtrl baseline.

**Strengths:**

The problem this method tackles is an important and impactful one. The manuscript is very clear and well written. The evaluation is adequate and the proposed method outperforms the shown baselines quantitatively. The qualitative results are certainly not as impressive as recent video generation models, like sora, but adequate given the limited capabilities of the SVD base model.

**Weaknesses:**

In my assessment, the main weakness of this submission is a lack of novel contributions. This paper is not the first to propose camera conditioning for text-to-video generation (e.g., line 93), it does not introduce epipolar attention, and it does not introduce Pucker coordinates. Similar to other works in this general area this is a "systems paper, which combines several known components into an adapter network. Systems papers are important and valuable, but only if they enable a new capability or solve a problem in some novel and better way. This is not the case here, because very similar systems have been described before, like MotionCtrl, and the quantitative improvements of this work are marginal (see e.g. Table 1). Without a clear technical contribution on the methods side and only marginal improvements for the targeted application, it seems difficult to get excited about this submission.

What I also find concerning is that the CameraCtrl paper is only tangentially mentioned. This paper has been on arxiv for more than 7 months and basically claims the same system's level contributions, including Plucker conditioning. CameraCtrl is not included in the baseline comparisons. The authors claim that it's concurrent work, but in an area that is so fast paced, I find it difficult to convince myself that a paper that has been online for such a long time should be treated as being concurrent. Regardless, this is not my primary concern.

**Questions:**

How does the proposed method compare with the CameraCtrl approach?

---

### Official Review · Reviewer_NugB · 2024-11-02

**Soundness:** 3
**Presentation:** 3
**Contribution:** 2
**Rating:** 5
**Confidence:** 4

**Summary:**

This paper presents a method to condition a pre-trained image to video model with explicit 3D camera control. The camera information is represented as plucker embeddings and features extracted from these are provided as input to the temporal layers. In addition, an epipolar attention layer is introduced to better preserve the geometry/rigidity of the reconstructed scenes. The method is trained on SVD using camera parameters obtained from both static and dynamic scenes.

**Strengths:**

- The presentation is clear.
- The ablation studies are good. The metrics used to evaluate the method are reasonable. A good set of visual results are provided in the supplementary.
- The design choices of using plucker embeddings and epipolar attention are reasonable.

**Weaknesses:**

- Many pieces of the work have been also discussed in other previous/concurrent work. Use of plucker embeddings for cameras is becoming a standard. The use of epipolar attention has been introduced in previous work that tackle multi-view generation. Hence, although reasonable, the paper does not introduce very specific novel contributions.
- While the use of epipolar attention improves scene rigidity, it is not discussed at all how this affects the dynamic videos. The epipolar constraints would not hold for objects that are moving.
- The dynamic video examples shown in the supplementary -seem to have relatively low motion.

**Questions:**

- As mentioned above, how do the epipolar constraints affect the dynamic video cases? (also related to the last comment above)
- When comparing to CameraCtrl (which seems closely related), it seems the text 2 video version of CameraCtrl is used, this paper also shows results on SVD. Are the comparisons done for both models using SVD?

---

### Official Review · Reviewer_wzan · 2024-11-02

**Soundness:** 3
**Presentation:** 3
**Contribution:** 2
**Rating:** 5
**Confidence:** 5

**Summary:**

This paper extends the I2V model SVD by adding camera control. The proposed method, CamCo, parameterizes input camera poses as Plücker coordinates and feeds the condition into temporal attention and the newly added Epipolar attention layers. These two layers are tuned to teach the network how to react to the provided 3D camera condition, while the remaining layers, e.g., self-attn. layers, are frozen to retain the quality of the generated videos.

**Strengths:**

The proposed method is explained clearly. The authors compare with a few baselines and show the best adherence to the input camera according to Table 1 (despite the most important one is missing). The curated dynamic video dataset can be a good contribution if released.

**Weaknesses:**

1. Despite the effort of annotating the dynamic dataset WebVid, from the results in the suppl. page, the foreground dynamic is still largely lost. Even the eagle example doesn't show prominent motion; in another example where a bird flying above a lake, the proposed method does produce more object translation than baselines, but the object size is fairly small. Arguably, the proposed method still suffers from the common problem shared with the state-of-the-art camera-conditioned methods, i.e., lost of foreground dynamics. Any idea how to improve here?
2. Limited novelty: Epipolar attention and Plücker coordinate are very standard in 3D generation field, e.g., [1]. Existing video generation methods have applied one of them, if not both, to facilitate camera conditioning, e.g., CameraCtrl also adopts Plücker coordinate to parameterize cameras. If the ideas are similar, I expect to see in-depth discussion/analysis why one is superior than the other. The submission nonetheless compares with CameraCtrl only "qualitatively" not quantitatively. CameraCtrl has released the code a while back and this submission follows its evaluation metrics, so quantitative comparison should not be too difficult.


[1] Kant et al., SPAD : Spatially Aware Multiview Diffusers, CVPR24. https://yashkant.github.io/spad/

**Questions:**

1. Why is epipolar attn. inserted before temporal attn., not after? Any intuition or empirical evidence?
2. Why training with WebVid doesn't result in a ShutterStock watermark? As far as I know, a big portion of WebVid videos contain a "ShutterStock" watermark. In fact, WebVid is also not publicly available anymore. Any method trained with the pre-downloaded copy (like the authors clarify in L546) is hard to reproduce. If any preprocessing is performed to prevent the watermark from emerging in the generated videos, the authors should disclose and explain the details.
3. Table 3 doesn't have a corresponding discussion. L.505-506 says Table 4, but I don't see Table 4 anywhere, so I assume it's a typo?
4. Will the curated Particle-SfM annotations for dynamic videos be released?

---

### Official Review · Reviewer_uLsi · 2024-11-03

**Soundness:** 3
**Presentation:** 3
**Contribution:** 3
**Rating:** 6
**Confidence:** 4

**Summary:**

The paper introduces CamCo, a novel framework for camera-controllable, 3D-consistent image-to-video generation, enabling fine-grained control over camera viewpoints while ensuring geometric consistency in the generated videos. This is achieved by: 1. camera pose parameterization with Plücker Coordinates; 2) epipolar attention to improve 3D consistency; 3) performing data curation and fine-tuning for dynamic Scenes using SFM. Results show a step forward comparing with prior camera controllable video generation in terms of accuracy of cameras and 3D consistency.

**Strengths:**

1. The authors implement a data curation pipeline that annotates in-the-wild videos with estimated camera poses using structure-from-motion algorithms. This enhances the model's ability to generate plausible object motion in addition to camera movements, addressing the challenge of synthesizing dynamic scenes.
2. The paper provides thorough quantitative and qualitative evaluations, demonstrating that CamCo outperforms baseline methods in terms of visual quality, camera controllability, and geometric consistency. Metrics like FID, FVD, and COLMAP error rates support these claims.
3. The inclusion of ablation studies validates the effectiveness of the proposed components, such as the Plücker coordinate parameterization and the ECA module. This strengthens the paper's contributions by showing the impact of each component.

**Weaknesses:**

1. The biggest weakness of the paper is its technical contribution, its main designs are Plücker coordinates and epipolar attention, but none of these is exactly novel, even on the constrained domain of camera controllable video generation --- the former was used in CameraControl [1] and the later was used in Collaborative Video Diffusion [2]. The authors should discuss what is novel about their approach while using these techniques.
2. Without sufficient dynamic training data, the model tends to overfit to static scenes with minimal object motion. Although the authors address this by curating additional dynamic data, it indicates a reliance on data quality and diversity --- this is especially concerning given that the authors choose do develop their model on SVD, a model that is known to generate very limited motions.
3. The quantitative evaluation primarily uses FID, FVD, and COLMAP error rates. While these are standard metrics, they may not fully capture perceptual quality, temporal coherence, or user satisfaction. For instance, FVD is known to be biased towards good qualitative single frames without taking the overall motion coherence into account. Incorporating additional metrics such as de-biased FVD [3] or user studies could provide a more comprehensive assessment of the model's performance.


[1] He et al. CameraCtrl: Enabling Camera Control for Text-to-Video Generation, in arXiv, 2024.

[2] Kuang et al. Collaborative Video Diffusion: Consistent Multi-video Generation with Camera Control, in NeurIPS, 2024.

[3] Ge et al., On the Content Bias in Fréchet Video Distance, in CVPR 2024.

**Questions:**

N/A

---

### Official Review · Reviewer_idaw · 2024-11-03

**Soundness:** 3
**Presentation:** 3
**Contribution:** 2
**Rating:** 5
**Confidence:** 4

**Summary:**

The paper proposes to address the image-to-video generation with precise camera control.  The author parameterizes the camera pose using Plücker coordinates and adopts a epipolar attention module to improve 3D consistency. The author further augments the training set with a curated dataset to better capture object movements. Experiments on different source datas demonstrate the effectiveness and generalization of the model.

**Strengths:**

**Good presentation**

The paper is easy to read and have good figures to show the method. Overall the pipeline is reasonable and each module is clear introduced.

**Good experimental results**

The model shows superior results over existing controllable camera image-to-video generation results, with much lower COLMAP error and FVD.

**Good generalization performance.**

I appreciate the author conduct rich experiments on multiple source unseen data to show the generalization ability. The author also shows the good 3D consistency in the generated videos.

**Weaknesses:**

**Very Limited Contribution**

The 2 fundamental modules introduced in this paper: Plücker embedding and epipolar attention are all commonly used techniques. For Plücker embedding, previous work in CameraCtrl (He et al., 2024)  also used the same technique. Even their used for text to video generation other than image-to-video, but the key techniques are the same, that is how to better incorporate the camera pose into the video generation model other than using R and t. Also works in 3D generation[1, 2] also uses the same one. Similar for epipolar attention used in (Tseng et al., 2023). Compare to Tseng, one improvement is the efficiency of the attention. However, I could not find experimental support to show the improvement of the speed or training time. Besides, the author doesn't compare to Tseng's attention, and I could not tell is there more significant difference with other implementations. Also for the dataset augmentation, it seems the pipeline directly follows MotionCtrl.


**Training set**

Given the author augments the training set, but in order to demonstrate the effectiveness of each proposed module, the comparison to MotionCtrl should be conducted on the same training set. The author should further mention this in the paper to ensure the improvement is coming from the better camera encoding and epipolar attention other than high-quality data sources.



**Object and camera movement decomposition**

MotionCtrl offers the flexibility to decompose the camera and object motions in generation. However, it seems the proposed method could not achieve this. Also the object motion shows in the paper and video seems very limited. I am expect to see more objects with higher dynamic motion for comparison.




[1] Chen et al. Ray Conditioning: Trading Photo-consistency for Photo-realism in Multi-view Image Generation. ICCV 2023
[2] kant et al. SPAD : Spatially Aware Multiview Diffusers. CVPR 204

**Questions:**

**MotionCtrl**

MotionCtrl offers user-specified object motions, and they show good object motions with custom trajectory. Why in the comparison section of the paper, the MotionCtrl always output very limited object motions.

---

### Meta-Review · Area_Chair_HgHW · 2024-12-20

**Metareview:**

Summary:
The paper tackles the important problem of camera-controlled image-to-video generation. It achieves this by using Plücker coordinates to parameterize camera pose input and an epipolar attention module to enforce epipolar constraints. With fine-tuning on real videos and estimated camera poses, the experiments show improved 3D consistency and camera trajectory control compared to prior methods.

Strength:
- The exposition is good. The paper is easy to read and the figures are well-prepared.
- Improved results over existing controllable camera image-to-video generation results

Weakness:
- Limited technical contributions.
- Foreground dynamics are limited (due to the use of epipolar constraints)

Justification:
- Four reviewers are leaning negative about the paper, primarily due to the limited contributions from the paper. Unfortunately, the authors did not engage with the reviewers in the rebuttal period. As the concerns from the reviewers are not resolved, the AC finds no ground to accept.

**Additional Comments On Reviewer Discussion:**

The authors did not provide rebuttal and answer questions from the reviewrs.

---

### Decision · Program_Chairs · 2025-01-22

Reject